# Peripheral blood transcriptome identifies high-risk benign and malignant breast lesions

Hong Hou[1]☯, Yali Lyu[2]☯, Jing Jiang[3], Min Wang[2], Ruirui Zhang[2], Choong-Chin Liew [4,5,6†], Binggao Wang[1]*, Changming Cheng[2]*

1 Qingdao Central Hospital/Qingdao Cancer Hospital, Qingdao, Shandong Province, People's Republic of China, 2 Huaxia Bangfu Technology Incorporated, Beijing, People's Republic of China, 3 Qingdao Lianchi Maternity and Infant Hospital, Qingdao, Shandong Province, People's Republic of China, 4 Golden Health Diagnostics Incorporated, Jiangsu, People's Republic of China, 5 Late of Department of Clinical Pathology and Laboratory Medicine, University of Toronto, Canada, 6 Harvard Medical School, Brigham and Women's Hospital, Boston, MA, United States of America

☯ These authors contributed equally to this work.
† Deceased.
* wbgqd1965@163.com (BW); cmcheng2005@163.com (CC)

**Data Availability Statement:** Data Availability Statement: All relevant data are within the manuscript and its supporting information. The gene expression profiles and the risk score

## Abstract

### Background

Peripheral blood transcriptome profiling is a potentially important tool for disease detection. We utilize this technique in a case-control study to identify candidate transcriptomic bio-markers able to differentiate women with breast lesions from normal controls.

### Methods

Whole blood samples were collected from 50 women with high-risk breast lesions, 57 with breast cancers and 44 controls (151 samples). Blood gene expression profiling was carried out using microarray hybridization. We identified blood gene expression signatures using AdaBoost, and constructed a predictive model differentiating breast lesions from controls. Model performance was then characterized by AUC sensitivity, specificity and accuracy. Biomarker biological processes and functions were analyzed for clues to the pathogenesis of breast lesions.

### Results

Ten gene biomarkers were identified (*YWHAQ, BCLAF1, WSB1, PBX2, DDIT4, LUC7L3, FKBP1A, APP, HERC2P2, FAM126B*). A ten-gene panel predictive model showed discriminatory power in the test set (sensitivity: 100%, specificity: 84.2%, accuracy: 93.5%, AUC: 0.99). These biomarkers were involved in apoptosis, TGF-beta signaling, adaptive immune system regulation, gene transcription and post-transcriptional protein modification.

### Conclusion

A promising method for the detection of breast lesions is reported. This study also sheds light on breast cancer/immune system interactions, providing clues to new targets for breast cancer immune therapy.

calculated by predictive model based on 10-gene panel were detailed listed in S1 and S2 Tables of Support Information.

**Funding:** Huaxia Bangfu Technology Incorporated [http://www.hxjdyl.com/en/gongsijieshao.html] sponsored this research. Changming Cheng, Yali Lyu, Min Wang, Ruirui Zhang are employees of Huaxia Bangfu Technology Inc. Choong-Chin Liew was a consultant of Huaxia Bangfu Technology Inc. The funder provided support in the form of salaries for authors [C. Cheng, Y. Lyu, M. Wang, R. Zhang], but did not have any additional role in the study design, data collection and analysis, decision to publish, or preparation of the manuscript. The specific roles of these authors are articulated in the 'author contributions' section.

**Competing interests:** The authors have read the journal's policy and have the following conflicts: Changming Cheng, Yali Lyu, Min Wang, Ruirui Zhang are employees of Huaxia Bangfu Technology Inc who sponsored this research. Choong-Chin Liew was a consultant of Huaxia Bangfu Technology Inc and the founder of Golden Health Diagnostics Incorporated. None of the other authors has any competing interests. There are no patents, products in development or marketed products to declare. This does not alter the authors' adherence to all the PLOS ONE policies on sharing data and materials.

## Introduction

Breast cancer is the most frequently diagnosed cancer and the leading cause of cancer death in women worldwide [1]. In recent years, the incidence of breast cancer in China has been increasing, and may eventually surpass incidence rates in developed countries [2]. According to the latest GLOBOCAN 2018 report, the age-standardized incidence of breast cancer per 100,000 population in China was 36.1, which is less than half that of the United States (84.9) and the United Kingdom (93.6), although the age-standardized mortality rates per 100,000 population do not differ appreciably between China at 8.8, America at 12.7 and the United Kingdom at 14.4 [3]. The relatively high death rate for breast cancer in China is mainly due to the rapid rise in the incidence of disease, whereas incidence is stable or decreasing in Western countries [4]. The annual percentage increase in breast cancer incidence from 1999 to 2008 is over 2% in urban China and is as high as 5.5% to 6.0% in rural China [5]. It has been predicted that the number of breast cancer patients in China in 2021 will approach 2.5 million in women aged 55–69 years [6]. In addition, a large proportion of breast cancer in China occurs in younger patients who are diagnosed at age less than 50 years, whereas the peak age of breast cancer onset has been approximately 70 years in America [7].

Breast cancer is regarded as potentially curable if diagnosed and managed at an early stage. Women diagnosed with early stage breast cancer (Stage I or II) have a better prognosis (5-year survival rate, 85–98%) than do those diagnosed with advanced breast cancer (5-year survival rate for Stage III or IV, 30–70%) [8]. In addition, according to the Breast Imaging Report and Data System (BI-RADS), breast lesions mammographically classified in Group 2 as definitely benign require no more treatment than do those identified during routine mammography screening. Lesions mammographically or ultrasonographically classified into Group 3 or higher, however, are recommended for shorter follow-up intervals or biopsy in view of their unclear potential for malignancy.

Breast lesions at an early stage are usually asymptomatic and undetectable by self-examination, resulting in delayed treatment. Currently, early detection of breast lesions is mainly dependent on mammography or ultrasound [9]. However the size, nodularity, and sensitivity of the breasts during lactation, makes imaging examination a challenge during this period [10]. Though mammography screening is helpful in reducing mortality from breast cancer [11], this method of detection is often ineffective, especially when the tumor is small. Furthermore, the false-positive and false-negative rates of mammography are relatively high for women with dense breast tissue, such as pre-menopausal women or those receiving menopausal hormone therapy [12]. Compared with mammography, ultrasound has advantages for women with dense breast tissue, but due to the poor resolution of this method in soft tissue, ultrasound is more suitable as a supplemental rather than a stand-alone screening method [13]. Thus novel, minimally invasive biomarkers have been sought to improve the early detection of breast lesions.

Blood is a "fluid connective tissue" [14], and blood cells continuously interact with tissue cells throughout the entire body. Therefore blood cells can act as "sentinels" that indicate health or the presence of disease [15]. Peripheral blood is frequently used in clinical research because it is easy to access and potentially carries information about disease status and physiological responses. We have previously reported [16] that peripheral blood transcriptome profiling has been applied in the screening and early detection of various non-hematologic disorders, including cancer [17–21].

In the present study, we compare the blood gene expression profiles in women with breast lesions and control women with no breast disease in order potentially to develop a non-invasive test for early stage breast cancer and breast lesions. The transcriptomic biomarkers of breast lesions were identified and the roles of these genes in biological processes and functions were analyzed for clues to the pathogenesis of breast lesions.

## Materials and methods

This study was approved by the Ethics Committee of the Qingdao Central (Tumor) Hospital (IRB no. KY-P201803601) on January 30[th] 2019. Participants were recruited to this study from January 31[st] 2019 to June 30[th] 2019. Sample acquisition was conducted between January 31[st] 2019 and June 30[th] 2019 at the Qingdao Central (Tumor) Hospital. 151 participants were enrolled, including 44 healthy controls and 107 patients with breast lesions (50 high risk lesions and 57 breast cancer). Written informed consent was obtained from all study participants and approved by the Ethics Committee of Qingdao Central (Tumor) Hospital. All authors in this manuscript had access to individual participants' information and medical records, and data was scrubbed after information collection.

A total of 107 blood samples from patients with breast lesions was obtained. The study population comprised 107 female adult patients (age range, 23–78 years; mean age: $50.6 \pm 11.2$ years), including 50 women with high-risk breast lesions and 57 breast cancer patients. All patients were recruited before they had undergone any form of treatment, including endocrinotherapy, radio/chemo-therapy, targeted therapy or surgery. The breast lesion cohorts were categorized according to pathological examination. All patients underwent mammography or ultrasound, and the results were analyzed and categorized according to the Breast Imaging Reporting and Data System (BI-RADS) Grades [22]. In cases where the grades of mammography and ultrasound were inconsistent, the higher grade was adopted. High-risk lesions were defined as BI-RADS Grades 3 to 5 with no evidence of cancer at biopsy.

### Blood collection, RNA isolation and RNA quality control

Blood samples (2.5 ml) were drawn using PaxGene Blood RNA tubes (PreAnalytix GmbH, Hombrechtikon, Switzerland) and total RNA was then isolated as described in a previous publication [11]. The integrity of the purified RNA was accessed by 2100 Bioanalyzer RNA 6000 Nano Chips (Agilent Technologies, Inc., Santa Clara, CA, USA) and the quantity of RNA was assessed by NanoDrop 1000 UV-Vis spectrophotometer (Thermo Fisher Scientific, Inc. Waltham, MA, USA). All RNA samples were assessed by RNA integrity number $\geq 7 \cdot 0$ and 28S:18S rRNA$\geq$1.0.

### Microarray hybridization and microarray data analysis

The gene expression profiles of all 151 samples, including 44 normal controls, 50 high-risk breast lesions and 57 breast cancer, were characterized by microarray hybridization as per the manufacturer's protocol (Gene Profiling Array cGMP U133 P2 [Affymetrix; Thermo Fisher Scientific, Inc.]). Blood total RNA (200 ng for each sample) was labeled and hybridized onto Affymetrix microarray according to the manufacturer's protocol. Gene expression profiles were accessed using Affymetrix Expression Console software (version 1.4.1; Affymetrix; Thermo Fisher Scientific, Inc.). The raw gene expression data were normalized using the MAS5 method to make it possible to compare the profiling variations among microarrays.

The data mining method utilized for this study mostly follows the strategy described in our previous report [23]. In brief, to identify gene biomarkers for distinguishing breast lesions (high-risk benign and cancer) from normal controls, the probe sets of interest were selected from the 54,675 probe sets on the Affymetrix Gene Profiling cGMP U133 P2 microarray, by filtration according to the following series criteria: the probe sets could be detected reliably ("present" call) in all the samples; the sets were present within the MAQC list as reported by MAQC Consortium; and the stably expressed probe sets, also deemed as internal reference genes, were removed. The microarray data was transformed by a logarithmic intensity to satisfy Gaussian distribution requirements. All sample data were randomly divided into a training set and a test set in a proportion of 7:3.

To accelerate the screening of breast lesion-specific gene expression signatures, an ensemble learning strategy called AdaBoost was executed. Instead of making restrictive assumptions regarding the training set as in traditional data mining methods, this boosting method first creates a set of weak classifiers by assigning them appropriate extra weights and then combines these weak classifiers into a strong classifier. AdaBoost has important and significant advantages in both accuracy and training time as compared with other data mining methods [24]. The transcriptomic features of the breast lesions were identified and used to construct the predictive model by AdaBoost. To classify the breast lesion group and the normal control group, the area under the receiver operating characteristic curve (AUC) sensitivity, specificity and accuracy were estimated in both the training and the test groups.

## Bioinformatics analysis

The GO and KEGG annotations of the selected transcriptomic genes were queried from the COXPRESdb v7 database [25]. The protein-protein interactions between each transcriptomic feature and its first neighbouring protein counterpart with number less than 20 were downloaded from the STRING database with a total confidence greater than or equal to 0.7. Gene-annotation enrichment analysis using the cluster Profiler R package was performed on signature genes and their correlative proteins. Gene Ontology (GO) terms were identified with a strict cutoff of adjusted $p < 0.05$ corrected with the Benjamini–Hochberg (BH) method and a false discovery rate (FDR) of less than 0.05. Reactome pathways were also identified, with a strict cutoff of $p < 0.05$ corrected with the BH method and a false discovery rate (FDR) of less than 0.05. The protein-protein interaction network and gene network with the final biomarkers was carried out with Cytoscape software.

## Results

For this study a total of 151 blood samples was collected, including 44 controls and 107 breast lesions (50 high-risk breast lesions and 57 breast cancer lesions). Patients with breast cancer were older than the controls and older than those with high-risk lesions. Most subjects in the control group were aged less than 60 years, whereas about half (49/107) of the patients in the breast lesion cohort were older than age 60 (Table 1). The BI-RADS Grades of patients in the breast lesion group are also summarized: for high-risk lesions, the number of lesions Grade 3 and 4 was similar; for breast cancer lesions, most of the patients were Grade 5 (Table 1).

The histopathology of the breast lesions is shown in Table 2. In the category of high-risk lesions, the main two types were hyperplasia-related disease and fibroadenoma. In the category of breast cancer, invasive breast cancer accounted for about 81% (46/57) of all histological types. Most of the samples were histological Grade II (26/40), 17 were unknown.

### Transcriptome profiling of peripheral blood samples from normal controls and breast lesions

Transcriptome profiling of peripheral blood samples taken from women in the two cohorts (normal controls 44, breast lesions 107), were generated using Affymetrix GeneChip U133Plus2.0. The profiles were then analyzed comparing breast lesions and normal control samples. A final ten transcriptomic gene biomarkers were identified (*YWHAQ*, *BCLAF1*, *WSB1*, *PBX2*, *DDIT4*, *LUC7L3*, *FKBP1A*, *APP*, *HERC2P2*, *FAM126B*) and were able to distinguish blood samples from patients with breast lesions from normal control samples. The corresponding gene symbols and fold changes of the final ten probe sets are listed in Table 3.

**Table 1. The basic characteristics of normal controls and breast lesions.**

|  | Normal controls | High risk lesions | Breast cancer |
|---|---|---|---|
| **Age(years)** |  |  |  |
| Min | 26 | 23 | 33 |
| Max | 6 | 68 | 78 |
| Mean | 42.6±11.6 | 44.9±10.0 | 55.6±9.6 |
| **Total-Age groups(years)** |  |  |  |
| 21–30 | 8 | 5 | 0 |
| 31–40 | 12 | 10 | 4 |
| 41–50 | 12 | 22 | 14 |
| 51–60 | 11 | 10 | 19 |
| 61–70 | 0 | 3 | 17 |
| 71–80 | 1 | 0 | 3 |
| Total | 44 | 50 | 57 |
| **BI-RADS Grades** |  |  |  |
| 3 |  | 23 | 0 |
| 4 |  | 27 | 10 |
| 5 |  | 0 | 42 |
| 6 |  | 0 | 5 |
| Total |  | 50 | 57 |

## Model selection and performance evaluation

Based on the ten candidate biomarkers we identified, a predictive model was constructed for discriminating breast lesions from normal controls using AdaBoost.

Fig 1 demonstrates using hierarchical cluster diagrams the performance of each single gene and the ten-gene panel for distinguishing breast lesions from controls for the entire 151 samples. The ten-gene panel exhibited a better performance than any of the single genes alone in clustering breast lesion samples from normal control samples.

To construct the predictive model, we divided the total data into a training set and a test set in proportions of 7:3. The predictive model built on the training set that contained a total of 105 samples included 80 breast lesions and 25 normal controls. The performance of the predictive model was then evaluated by the completely independent samples in the test set, which contained a total of 46 samples, including 27 breast lesions and 19 normal controls. The performances of the training set and the test set are shown in Table 4. In terms of specificity and accuracy both training set and test set performed well; the test set sensitivity was 100%, and specificity and accuracy were 84.2% and 93.5%, respectively. Three of the 19 normal control samples in the test set were predicted as positive results; the reason for these false-positive results requires further study in a larger cohort. The ten-gene biomarker panel also exhibited a higher ROC AUC as compared with any single biomarker, in both the training set and the test set, as shown in Fig 2. As shown in Fig 3, the box-whisker plot illustrates the well-separated distribution of prediction scores of breast lesions and normal controls, based on the 10-gene panel and AdaBoost algorithm.

## Protein networks and functional enrichment analysis

The proteins interacting with the ten candidate biomarkers used for the model construction were downloaded from the STRING database, and a total of 147 proteins were identified with a confidence greater or equal to 0.7. The detailed interaction of these proteins is shown in Fig 4. Functional enrichment analysis was conducted and pathways were identified with a strict

**Table 2. The histopathological types of breast lesions.**

| Diagnosis | Subtype/ Histological grade | Number of samples |
|---|---|---|
| **High risk lesion (50)** | Hyperplasia | 21 |
| | Fibroadenoma | 17 |
| | Papilloma | 6 |
| | Phyllode tumor | 3 |
| | Adenolipoma | 1 |
| | Mammary duct ectasia | 1 |
| | Lobular atrophy | 1 |
| **Invasive breast cancer (46)** | Histological grade I | 2 |
| | Histological grade II | 18 |
| | Histological grade III | 9 |
| | Histological grade unknown | 17 |
| **Ductal carcinoma in situ (3)** | Histological grade I | 0 |
| | Histological grade II | 2 |
| | Histological grade III | 1 |
| **Papillary breast cancer (2)** | Histological grade I | 0 |
| | Histological grade II | 2 |
| | Histological grade III | 0 |
| **Invasive lobular carcinoma (2)** | Histological grade I | 0 |
| | Histological grade II | 2 |
| | Histological grade III | 0 |
| **Squamous cell carcinoma (2)** | Histological grade I | 0 |
| | Histological grade II | 1 |
| | Histological grade III | 1 |
| **Tubular carcinoma (1)** | Histological grade I | 1 |
| | Histological grade II | 0 |
| | Histological grade III | 0 |
| **Mucinous carcinoma of breast (1)** | Histological grade I | 0 |
| | Histological grade II | 1 |
| | Histological grade III | 0 |
| **Total samples** | | **107** |

cutoff of adjusted p<0.05, corrected with the Benjamini–Hochberg (BH) method. Our analysis identified 53 pathways consisting of these ten transcriptomic gene biomarkers, and we chose for further analysis the top 16 pathways with the highest p-adjusted values. As indicated in Fig 5A, these pathways were mainly involved in apoptosis, TGF-beta signaling, adaptive immune system regulation, gene transcription and post-transcriptional protein modification. The relationship of the transcriptomic gene biomarkers identified and the pathways involved are indicated in Fig 5B.

## Discussion

In this study we report a method for differentiating breast lesions—including high-risk benign breast lesions and malignant breast lesions—from normal controls using blood transcriptomic gene expression analysis. We collected blood samples from healthy control women with no breast disease and from breast lesion patients, and focused on identifying blood transcriptomic features that can distinguish the two groups. We identified ten genes that can detect breast lesions with an accuracy higher than 90%. These preliminary results are encouraging, but further research is needed for validation.

**Table 3. Candidate biomarkers for distinguishing breast lesions from controls.**

| Probe set ID | Gene Symbol | Gene Title | Fold Change | Regulation |
|---|---|---|---|---|
| 202887_s_at | DDIT4 | DNA damage inducible transcript 4 | 2.0014469 | up |
| 214953_s_at | APP | amyloid beta (A4) precursor protein | 1.97339 | up |
| 214119_s_at | FKBP1A | FK506 binding protein 1A | 1.8358978 | up |
| 202876_s_at | PBX2 | pre-B-cell leukemia homeobox 2 | 1.7287292 | up |
| 200693_at | YWHAQ | tyrosine 3-monooxygenase/ tryptophan 5-monooxygenase activation protein, theta | 1.0801506 | up |
| 217317_s_at | HERC2P2 | hect domain and RLD 2 pseudogene 2 | -1.2864129 | down |
| 208835_s_at | LUC7L3 | LUC7-like 3 pre-mRNA splicing factor | -1.3374902 | down |
| 201296_s_at | WSB1 | WD repeat and SOCS box containing 1 | -1.350631 | down |
| 201101_s_at | BCLAF1 | BCL2-associated transcription factor 1 | -1.4449192 | down |
| 1554178_a_at | FAM126B | family with sequence similarity 126, member B | -1.4458523 | *down* |

As breast cancer is the leading cause of cancer death in women, early detection has played a critical role in the management of this disease, especially for those many women whose breast cancer has no symptoms [26]. High-risk breast lesions represent a group of lesions, which clinically, morphologically, and biologically heterogeneous carry an increased risk of breast cancer, albeit to various degrees [27]. The threat of high-risk though benign breast lesions should not be underestimated. High-risk breast lesions convey a high relative risk for a later breast cancer with a cumulative incidence of 29% within 25 years [28–30]. Since high risk lesions are frequently also asymptomatic, we should explore new strategies for the detection of all breast lesions, including both breast cancer and high risk lesions not yet malignant.

In current clinical practice the most common tool used for the early detection of breast lesions is mammographic screening with complementary ultrasound. Definitive diagnosis requires biopsy. Since mammography carries high false positive rates and biopsy is traumatically invasive, the development of a novel, sensitive, non-invasive approach for early detection of breast lesions is essential to complement existing methods of detection.

To develop such an approach, we have utilized methods for cancer detection described in our blood transcriptome study 'and our previous reports [17,31, 32], and identified a ten-gene panel (Table 3) from peripheral blood gene expression profiles. The predictive model we developed based in the ten-gene panel performed well both in the training set and test set (Figs 1 and 2). In the independent test set, the ten-gene panel differentiated breast lesions from normal controls with sensitivity of 100%, specificity of 84.2%, accuracy of 93.5% (Table 4). We are planning to follow these patients over the next few years to confirm whether those 3 false positive samples are true negative samples. Since it is essential to predict breast lesions at early stages for prevention and optimal treatment, we are interested to know whether the biomarkers identified in the present retrospective study are effective in predicting high-risk lesions or breast cancer. We also expect to further evaluate the blood based biomarkers in a future prospective study.

Among the ten candidate biomarkers we identified (YWHAQ, BCLAF1,WSB1, PBX2, DDIT4, LUC7L3, FKBP1A, APP,HERC2P2,FAM126B), five genes (DDIT4, APP, FKBP1A, PBX2, YWHAQ) were upregulated in breast lesion patients as compared with normal controls, and the other five genes were downregulated (FAM126B, BCLAF1, WSB1, LUC7L33, HERC2P2.) There were a total of 147 proteins interacting with the ten transcriptomic genes (Fig 4), and functional enrichment analysis of these proteins showed they were mainly associated with apoptosis, TGF-beta signaling, adaptive immune system regulation, gene transcription and post-transcriptional protein modification (Fig 5). The gene involved in apoptosis was YWHAQ and the gene involved in TGF-beta signaling was FKBP1A. YWHAQ also joined the process of gene transcription with DDIT4. In adaptive immune system regulation, FKBP1A

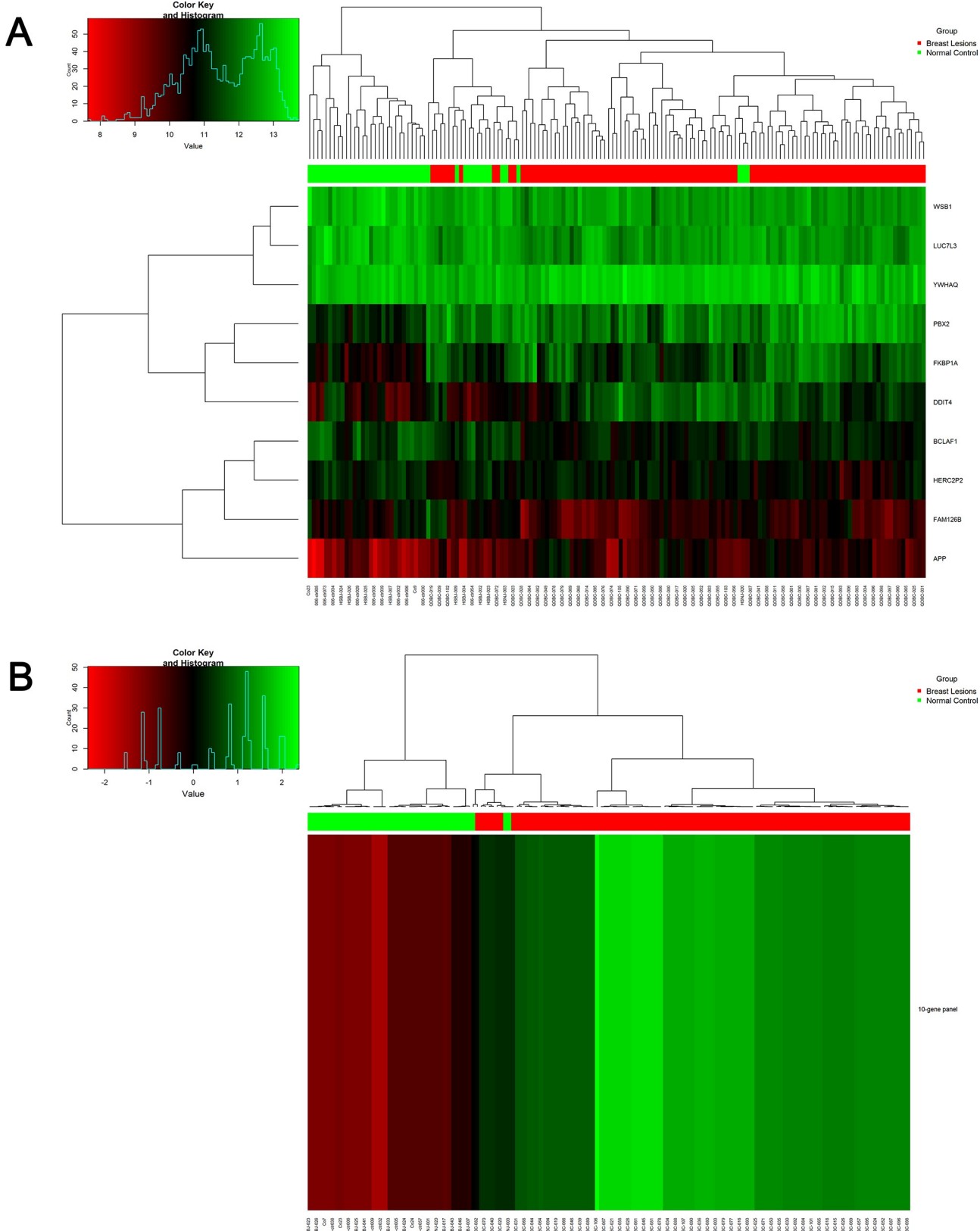

**Fig 1.** Heat map of gene expression and hierarchical cluster diagram showing 10 single candidate genes (A) and 10-gene combination (B) for clustering the 151 samples including 107 breast lesions and 44 normal controls. Dendrogram generated using "Heatmap" function in R with default settings.

**Table 4. Model construction and performance evaluation.**

|  | Training set | | Test set | |
|---|---|---|---|---|
|  | **Breast lesions** | **Normal Control** | **Breast lesions** | **Normal Control** |
| Positive | 80 | 0 | 27 | 3 |
| Negative | 0 | 25 | 0 | 16 |
| Total | 80 | 25 | 27 | 19 |
| Sensitivity | 100% | | 100% | |
| Specificity | 100% | | 84.2% | |
| Accuracy | 100% | | 93.5% | |
| ROC AUC | 1 | | 0.99 | |

participates in the calcineurin activation of *NFAT* and *WSB1* and plays a role in antigen processing involving ubiquitination and proteasome degradation. *WSB1* is also involved in the post-transcriptional protein modification process, neddylation.

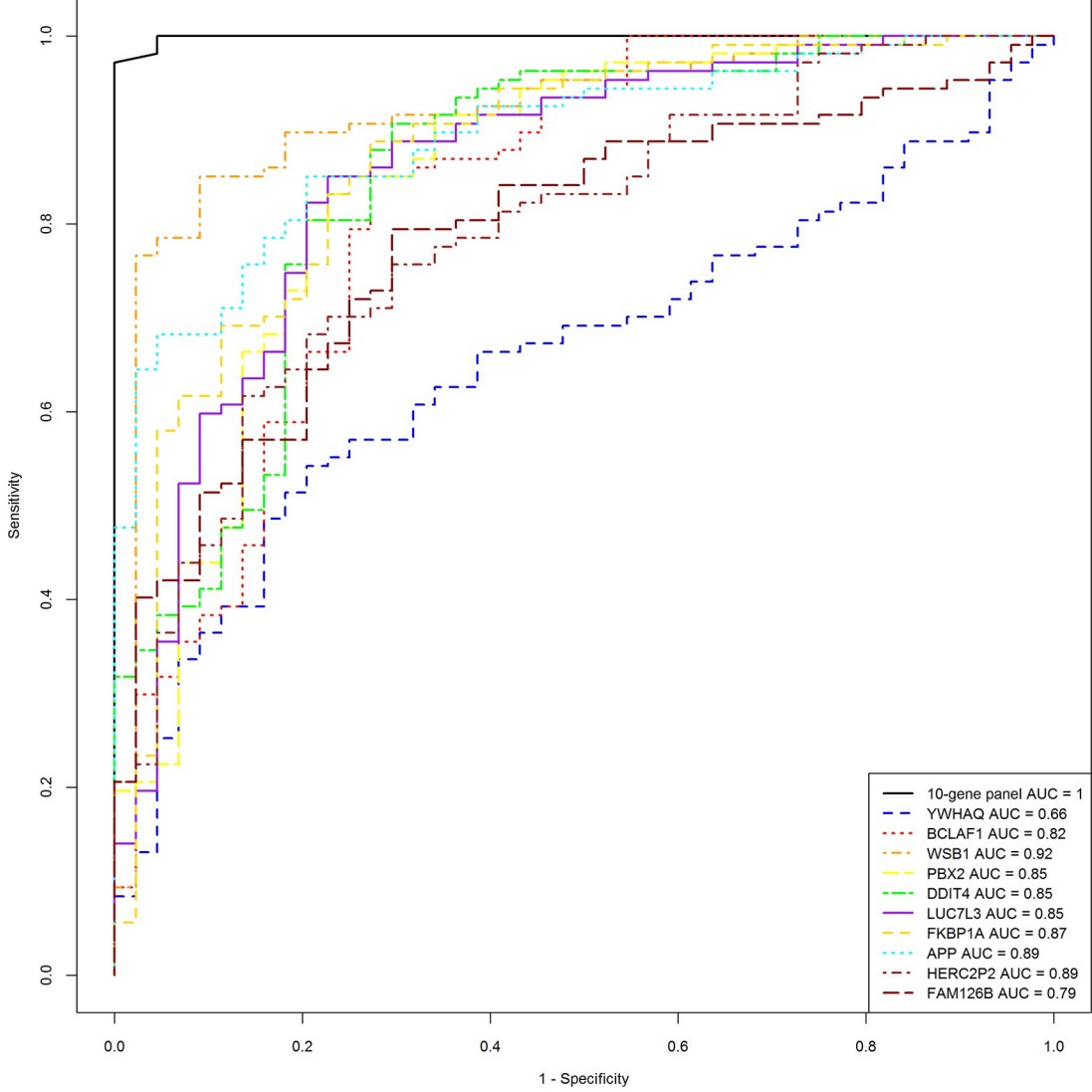

**Fig 2. ROC curve analysis for comparison of breast lesions versus normal controls.**

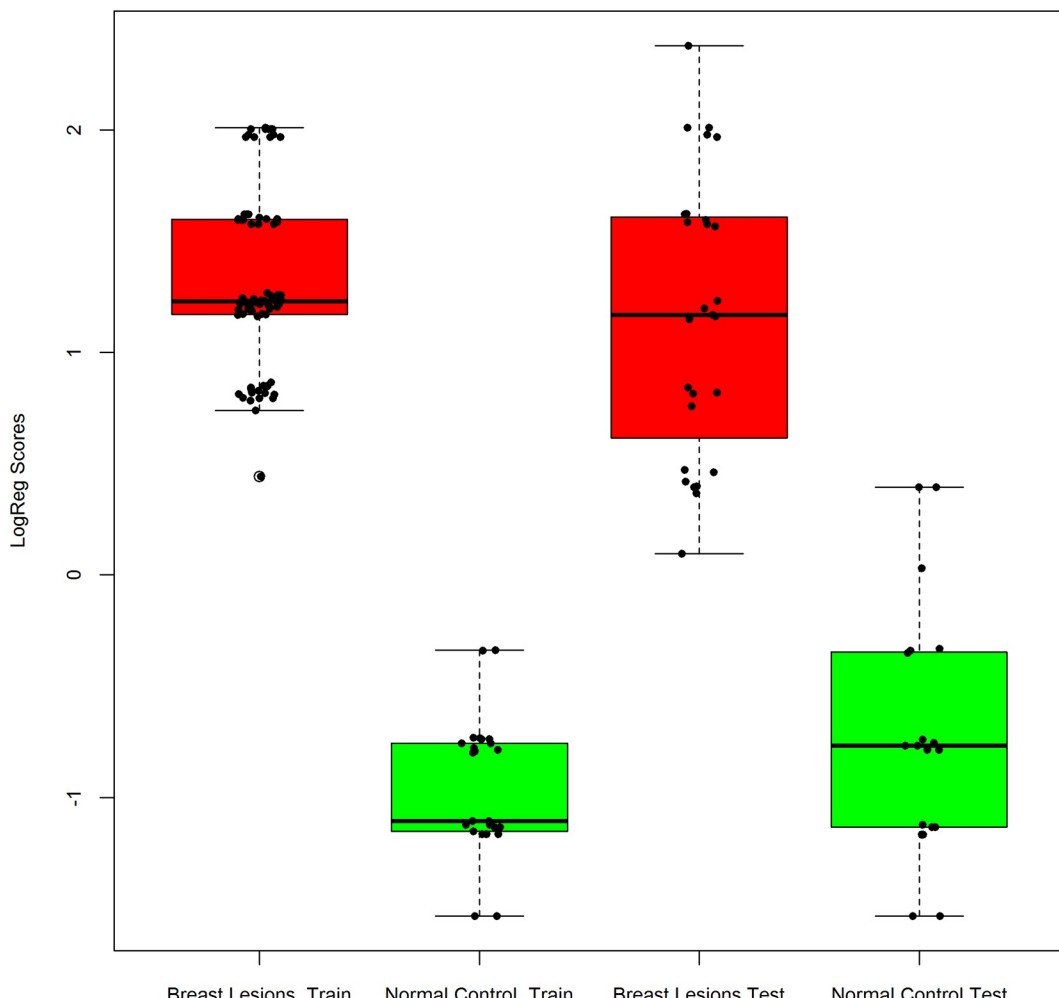

**Fig 3. Box-whisker plot to display the decision scores in breast lesions and normal controls in the training set and test set.** Red, breast lesions, Green, normal control.

The most over-expressed biomarker in the breast lesion group was *DDIT4* (for DNA-damage-inducible transcript 4), also known as *REDD1* or *RTP801*. The major function of the protein encoded by *DDIT4* is to inhibit *mTORC1*, which is induced by various stress stimulus in the hypoxia inducible factor (HIF) family [33,34]. Pinto et al reported that high levels of *DDIT4* were significantly associated with a worse prognosis (recurrence-free survival, time to progression and overall survival) in several cancer types, including breast cancer [35]. Their previous work indicated that high *DDIT4* expression was also an independent factor for a shorter disease-free survival in chemotherapy-resistant triple negative breast tumors [36]. In another report, the dysregulation of basal *DDIT4* gene expression in several cancer types (e.g. lung, breast, prostate) can be altered by promyelocytic leukemia (PML) and lead to mTOR activation and cancer progression [37]. *DDIT4* also acts as a pro-death transcript in the calcitriol inducing endoplasmic reticulum -stress-like response in breast cancer [38]. Consistent with these reports, in our study *DDIT4* was also upregulated in breast lesions, therefore it might serve as a novel prognostic biomarker and is a potential candidate for the development of targeted therapy for breast cancer.

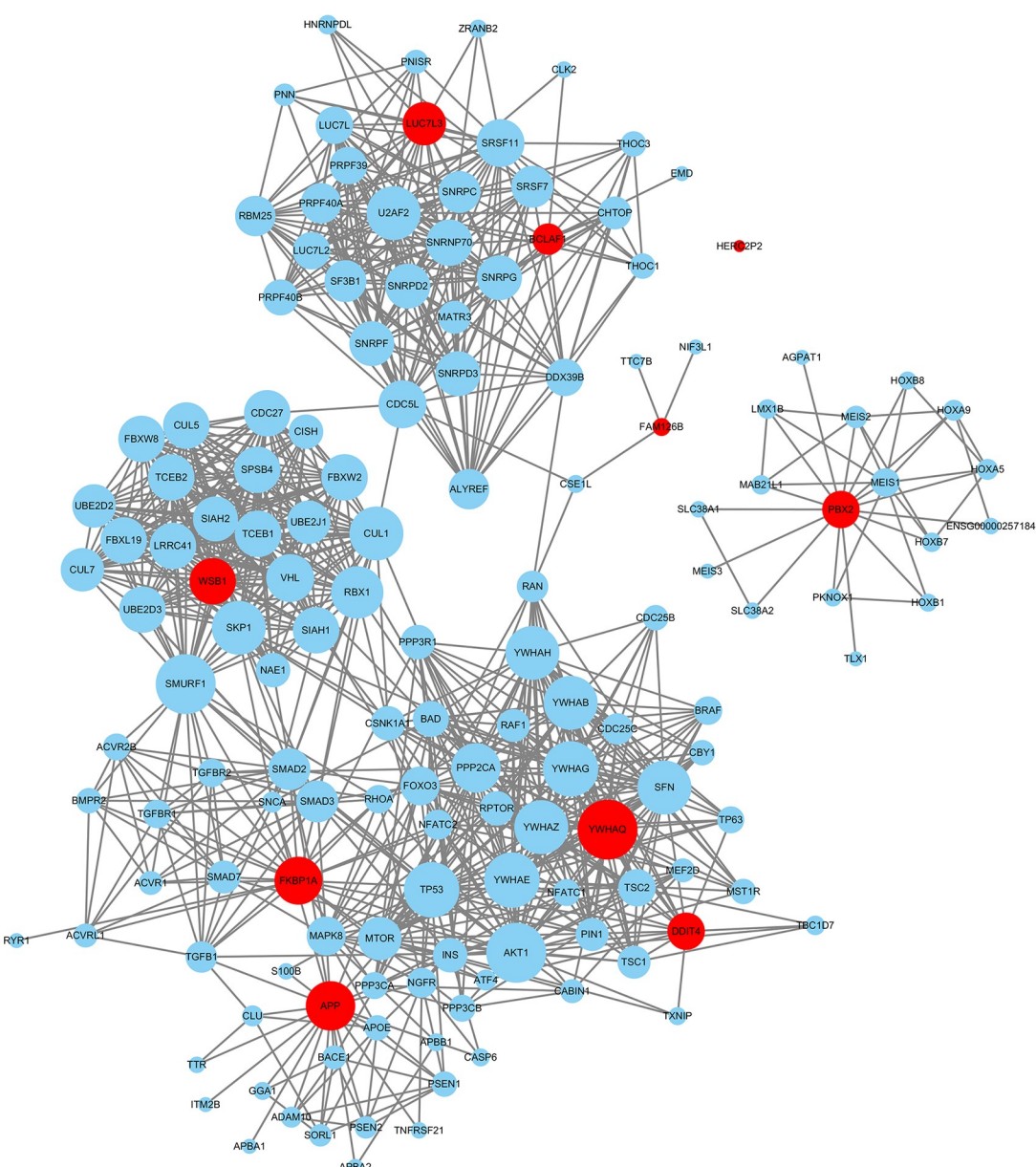

**Fig 4.** Interaction map of 10 transcriptomic gene biomarkers (red circles) and their interacting proteins (blue circles), using the edge weight cutoff 0.7 (total confidence greater or equal to 0.7).

Another upregulated gene, *YWHAQ* encodes the 14-3-3 proteins, which belong to a group of highly conserved proteins that are essential components of key signaling pathways involved in apoptosis and cell proliferation. These proteins interact with proteins such as Raf, BAD, protein kinase C (PKC), and phosphatidylinositol 3-kinase [39]. The products of *YWHAQ* (14-3-3ε) regulate TP53 through protein-protein interactions and post-translational modifications [40], and the germline variation in the TP53 network genes *PRKAG2*, *PPP2R2B*, *CCNG1*, *PIAS1* and *YWHAQ*, might affect prognosis and treatment outcome in breast cancer patients [41]. TP53 is closely associated with breast cancer; women who have germline TP53 mutations have a very high risk of breast cancer of up to 85% by age 60 [42]. Combining these reports

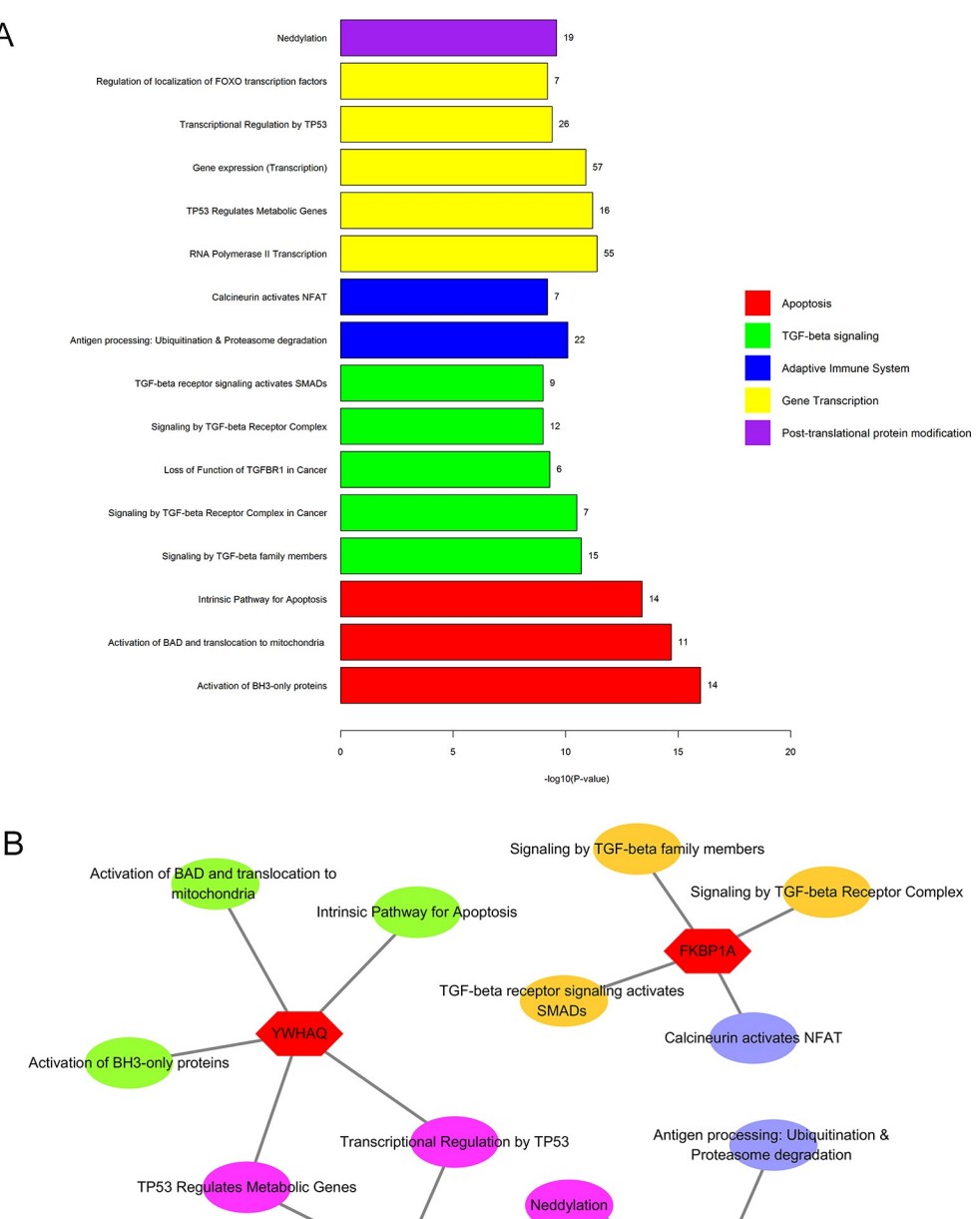

**Fig 5. Functional categorization of transcriptomic gene biomarker-related genes.** A: The top 16 pathways containing the 10 transcriptomic gene biomarkers. B: The relationship of the engaged transcriptomic genes and the pathways involved.

with our results suggests the TP53 network gene *YWHAQ* may act as a predictor and new therapy target for breast cancer.

In the present study, *FKBP1A* participated in both the TGF-beta signaling and calcineurin activation of *NFAT*. *FKBP1A*, also named *FKBP12*, is a member of the FK-506-binding protein (*FKBP*) family, and its expression in cells is ubiquitous [43, 44]. *FKBP1A* mediates the immunosuppressive and antitumor effects of rapamycin [45], widely used in the treatment of breast cancer [46, 47]. One study on Eph receptors and invasive breast carcinoma suggested that the

level of *FKBP1A* was significantly affected by *EphB6*, which was a target mRNA of miR-100, the changes in miRNAs and the target mRNA may have a role in *PI3K/Akt/mTOR* pathways [48]. *FKBP1A* has also been shown to inhibit TGF-beta type 1 receptor [49] and it was found overexpressed in childhood astrocytomas, which presented as the *EGFR/FKBP12/HIF-2alpha* pathway [50]. While an aberration of TGF-beta type 1 receptor is associated with a significantly increased risk of breast cancer [51], *FKBP1A* may also be associated with an elevated risk of breast cancer, as our study indicated.

Among the downregulated genes, *WSB1* is associated with antigen processing, specifically: ubiquitination and proteasome degradation and the post-transcriptional protein modification process, neddylation. *WSB-1* (WD-40 repeat-containing SOCS Box protein), is the substrate recognition element of an Elongin Cullin SOCS (ECS box) E3 ubiquitin ligase complex [52] and it was identified as a transcriptional target of HIF [53]. In the only study on the role of WSB1 in breast cancer, Poujade et al found that *WSB-1* plays an important role in breast cancer metastasis. By knocking down the *WSB-1* gene in breast cancer cell lines, these investigators found that the downregulation of *WSB-1* gene expression levels could significantly decrease the metastatic potential of breast cancer [54].

Our results were inconsistent with the above report, however, since *WSB1* was decreased in our breast lesion group. The role of *WSB1* in other types of cancer is also controversial; this gene was involved in pancreatic cancer progression [55] and metastatic potential of osteosarcoma [53], but its high expression was associated with good prognosis and favorable outcome of neuroblastoma [56]. So the definite function of *WSB1* in breast cancer remains unclear.

The gene mutations related to carcinogenesis, such as p53, BRCA1 / BRCA2, have been widely observed in breast tumor cells; however they have not been identified in our study with significant expression variation between breast lesion and healthy control group in peripheral blood. There are several possible reasons for this. Although tumor cells could be released into a patient's peripheral blood, the proportion of such cells as compared with white blood cells would be very low, even for patients with advanced disease. White blood cells predominate in the cell spectrum of peripheral blood, and therefore blood gene expression signatures would largely reflect these abundant blood white cells rather than the rare circulating tumor cells. In addition, as blood white cells and tumor cells play different biological roles in the process of carcinogenesis their gene expression profiles also differ. Gene expression variations in blood white cells, for example, more likely reflect interactions between the immune system and the tumor rather than reflecting intrinsic changes within the tumor cells themselves. These differences might be an important reason why the driver genes that have been observed in tumor cells did not show abnormal signals in the gene expression profile of peripheral blood in this study. Further study is required in order to identify the signaling pathways of blood cells and their interaction with cancer cells to better understand the roles of blood cells in carcinogenesis.

Our study has several limitations. First, the sample size was relatively small and different genes or more genes that have better discriminatory power may be validated among a larger independent cohort of patients. For example, our samples show some age variation among the healthy controls, the women with high risk lesions and the breast cancer patients. Age has been regarded as an important risk factor for cancer, as the incidence of most cancers increases with age. In this study, which is restricted by a limited sample size, we tried to optimize the algorithm to eliminate the interference of age factors as much as possible. However, it is hard to confirm that the biomarkers derived are completely unrelated to age.

We intend to confirm the effectiveness of our data mining method in further studies, using a larger sample size with age-matched patients. Second, the nature of the mechanisms driving the different transcriptomic biomarkers in peripheral blood is not yet clear, and the function

of some biomarkers requires further study. We are currently exploring the expression differences of the ten candidate biomarkers between high-risk breast lesions and breast cancer, which study may be helpful for the differential diagnosis of high risk lesions and breast cancer.

Finally, RNA sequencing (RNAseq) has been proven an efficient tool for transcriptome analysis, especially for exploring expression signatures of unknown transcript fragments and revealing the signaling pathways beneath, An interesting subject for future study would be to compare the variations in gene expression signatures between RNAseq and the microarray method.

Using peripheral blood gene expression profiles we identified ten transcriptomic biomarkers that could distinguish women with high-risk breast lesions and breast cancer from normal controls. Our model, based in the ten transcriptomic biomarkers identified, has shown good discriminatory power between breast lesion and control subjects. Our functional enrichment analysis suggested that our candidate biomarkers were mainly involved in apoptosis, TGF-beta signaling, adaptive immune system regulation, gene transcription and post-transcriptional protein modification. This study has therefore established a promising methodology for the non-invasive detection of breast lesions, and we have also shed light on the pathogenic mechanisms of breast cancer and provided clues to new targets for breast cancer therapy, especially therapies related to immune treatment.

## Supporting information

**S1 Checklist.** *PLOS ONE* **clinical studies checklist.**
(DOCX)

**S2 Checklist. STROBE statement—checklist of items that should be included in reports of observational studies.**
(DOCX)

**S1 Table. Blood-based gene expression profiles.**
(XLSX)

**S2 Table. Risk scores of samples.**
(XLSX)

## Acknowledgments

The authors would like to thank Qian Shi, who performed the Affymetrix microarray studies and Isolde Prince, who helped with the editing of the manuscript.

## Author Contributions

**Conceptualization:** Hong Hou, Yali Lyu, Choong-Chin Liew, Binggao Wang, Changming Cheng.

**Data curation:** Hong Hou, Jing Jiang, Min Wang, Binggao Wang.

**Formal analysis:** Min Wang, Ruirui Zhang.

**Funding acquisition:** Changming Cheng.

**Methodology:** Ruirui Zhang.

**Project administration:** Yali Lyu.

**Resources:** Hong Hou, Jing Jiang, Binggao Wang.

**Software:** Min Wang.

**Supervision:** Binggao Wang, Changming Cheng.

**Visualization:** Ruirui Zhang.

**Writing – original draft:** Yali Lyu.

**Writing – review & editing:** Choong-Chin Liew, Changming Cheng.

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
