## [Decision Letter · Decision Letter 0]

30 Jan 2020

PONE-D-19-31191

Peripheral blood transcriptome identifies high-risk benign and malignant breast lesions

PLOS ONE

Dear Dr Liew,

Thank you for submitting your manuscript to PLOS ONE. After careful consideration, we feel that it has merit but does not fully meet PLOS ONE’s publication criteria as it currently stands. Therefore, we invite you to submit a revised version of the manuscript that addresses the points raised during the review process.

We would appreciate receiving your revised manuscript by Mar 15 2020 11:59PM. To enhance the reproducibility of your results, we recommend that if applicable you deposit your laboratory protocols in protocols.io, where a protocol can be assigned its own identifier (DOI) such that it can be cited independently in the future. For instructions see: http://journals.plos.org/plosone/s/submission-guidelines#loc-laboratory-protocols

We look forward to receiving your revised manuscript.

Kind regards,

Sumitra Deb, PhD

Academic Editor

PLOS ONE

2. We noticed you have some minor occurrence(s) of overlapping text with the following previous publication(s), which needs to be addressed:

https://doi.org/10.3892/ol.2018.8577

https://doi.org/10.1038/s41416-018-0056-3

In your revision ensure you cite all your sources (including your own works), and quote or rephrase any duplicated text outside the Methods section. Further consideration is dependent on these concerns being addressed.

"Funder: Huaxia Bangfu Technology Inc

https://www.hxjdyl.com/en/gongsijieshao

We note that one or more of the authors are employed by commercial companies: Huaxia Bangfu Technology Inc and Golden Health Diagnostics Incorporated.

Reviewers' comments:

Reviewer's Responses to Questions

**Comments to the Author**

1. Is the manuscript technically sound, and do the data support the conclusions?

Reviewer #1: Partly

Reviewer #2: Yes

Reviewer #3: No

2. Has the statistical analysis been performed appropriately and rigorously? 

Reviewer #1: Yes

Reviewer #2: Yes

Reviewer #3: No

3. Have the authors made all data underlying the findings in their manuscript fully available?

Reviewer #1: No

Reviewer #2: Yes

Reviewer #3: Yes

4. Is the manuscript presented in an intelligible fashion and written in standard English?

Reviewer #1: Yes

Reviewer #2: Yes

Reviewer #3: Yes

5. Review Comments to the Author

Reviewer #1: In this manuscript authors have reported a method for detection of breast lesions using peripheral blood transcriptomic profiling. Overall the manuscript has been designed well but some issues need to be addressed. Minor: The font in Figure 1 should be made clearer. It was a little hard to read. Major: Definitely a larger sample size and more validation studies will be needed to support this method. Did authors try RNAseq instead of microarray? They need to address the fact that microarray has a limitation in which it does not allow accurate assessment of low signal intensities. And may also give background hybridization. Also the authors need to provide information on the ER, HER2 status of the lesions if available.

Reviewer #2: The authors have described a ten-gene panel signature that can be used as a predictive model to detect high risk breast lesions as well as breast cancer. The authors have performed the research very comprehensively and all the analyses performed is scientifically sound. While I do believe the research performed is of significant importance in the field, I do request the authors to address a few concerns before the manuscript can be accepted for publication:

1) On page 9, the authors mention that patients with breast cancer were older than the controls as well as the ones with high-risk lesions. How do the authors know that the ten gene panel signature isn't simply reflective of aging and is directly related to the process of carcinogensis.

2) Did the author check for the known drivers of cancer and in particular breast cancer such as BRCA1,BRCA2, p53 and such other genes? Or was the analysis done in a way to find out the other genes beyond the ones that were already identified in the literature?

3) Is there a way to follow up to see if the 3 'false-positive' normal samples in their test set go on to develop high risk breast lesions and therefore check if the ten-gene panel signature is actually predictive of the early stage carcinogenesis/development of high-risk lesions?

4) While the premise of the project and that of the manuscript is to discover a ten-gene signature panel that is predictive of developing high risk breast lesions, all the samples analyzed are grade 3 or 4 (BI-RADS). Can authors actually test if this panel is actually predictive of development of high-risk lesions by testing their gene signature panel in the grade I or grade II patients?

5) Have the authors tested to see if among the 147 protein interacting partners, the transcripts of the interacting partners were also altered in their RNAseq analyses? This is only to understand if multiple genes in the pathways described in the manuscript have been altered, and therefore narrow down a potential pathway of interest for further studies?

Once these comments are addressed, the manuscript can be considered for publication.

Reviewer #3: Comments:

1. It is not clear from the description about the parameters used to identify the 10 candidate genes.

2. It would be interesting to know the comparison of the various gene expression profile between the high risk benign breast lesion vs malignant breast lesions

3. Since the sample size is small it is important to use an age matched control.

4. The false positivity observed in 3 of the 19 controls of the predictive model is not justified despite of small sample size

5. Significant figures should be considered while calculating Fold change.

6. PLOS authors have the option to publish the peer review history of their article (what does this mean?). If published, this will include your full peer review and any attached files.

Reviewer #1: No

Reviewer #2: No

Reviewer #3: No

---

## [Author Response · Author response to Decision Letter 0]

23 Feb 2020

Response: We hope we have met PLOS ONE style requirements.

2. We noticed you have some minor occurrence(s) of overlapping text with the following previous publication(s), which needs to be addressed:

https://doi.org/10.3892/ol.2018.8577

https://doi.org/10.1038/s41416-018-0056-3

In your revision ensure you cite all your sources (including your own works), and quote or rephrase any duplicated text outside the Methods section. Further consideration is dependent on these concerns being addressed.

Response: the duplicated text with previous publications has been rephrased on page 6/7 and page 20. 

"Funder: Huaxia Bangfu Technology Inc

https://www.hxjdyl.com/en/gongsijieshao

We note that one or more of the authors are employed by commercial companies: Huaxia Bangfu Technology Inc and Golden Health Diagnostics Incorporated.

Response: We have added the following statements to our Cover Letter:

Funding Statement

Huaxia Bangfu Technology Incorporated [http://www.hxjdyl.com/en/gongsijieshao.html] sponsored this research. Changming Cheng, Yali Lyu, Min Wang, Ruirui Zhang are employees of Huaxia Bangfu Technology Inc. Choong-Chin Liew was a consultant of Huaxia Bangfu Technology Inc. The funder provided support in the form of salaries for authors [C. Cheng, Y. Lyu, M. Wang, R. Zhang], but did not have any additional role in the study design, data collection and analysis, decision to publish, or preparation of the manuscript. The specific roles of these authors are articulated in the ‘author contributions’ section.

Competing Interests Statement

The authors have read the journal’s policy and have the following conflicts: Changming Cheng, Yali Lyu, Min Wang, Ruirui Zhang are employees of Huaxia Bangfu Technology Inc who sponsored this research. Choong-Chin Liew was a consultant of Huaxia Bangfu Technology Inc and the founder of Golden Health Diagnostics Incorporated. None of the other authors has any competing interests. There are no patents, products in development or marketed products to declare. This does not alter the authors’ adherence to all the PLOS ONE policies on sharing data and materials.

Author contributions

Hong Hou Conceptualization, Data curation, Resources 

Yali Lyu Conceptualization, Project administration, Writing - original draft, 

Jing Jiang Data curation, Resources

Min Wang Data curation, Formal analysis, Software 

Ruirui Zhang Formal analysis, Methodology, Visualization 

Changming Cheng Conceptualization, Supervision, Writing - review & editing, Funding acquisition

Choong-Chin Liew Conceptualization, Writing - review & editing

Binggao Wang Conceptualization, Data curation, Resources, Supervision

Reviewers' comments:

Reviewer's Responses to Questions

Comments to the Author

1. Is the manuscript technically sound, and do the data support the conclusions?

Reviewer #1: Partly

Reviewer #2: Yes

Reviewer #3: No

2. Has the statistical analysis been performed appropriately and rigorously? 

Reviewer #1: Yes

Reviewer #2: Yes

Reviewer #3: No

3. Have the authors made all data underlying the findings in their manuscript fully available?

Data Availability Statement: All relevant data are within the manuscript and its supporting information. The gene expression profiles and the risk score calculated by predictive model based on 10-gene panel were detailed listed in S1 and S2 Tables of Support Information.

Reviewer #1: No

Reviewer #2: Yes

Reviewer #3: Yes

4. Is the manuscript presented in an intelligible fashion and written in standard English?

Reviewer #1: Yes

Reviewer #2: Yes

Reviewer #3: Yes

5. Review Comments to the Author

Reviewer #1: In this manuscript authors have reported a method for detection of breast lesions using peripheral blood transcriptomic profiling. Overall the manuscript has been designed well but some issues need to be addressed. Minor: The font in Figure 1 should be made clearer. It was a little hard to read. 

Response: Figure 1 was rephrased on page 13 and I hope that the font is clearer now. 

Fig. 1. Heat map of gene expression and hierarchical cluster diagram showing 10 single candidate genes (A) and a 10-gene combination (B) for clustering the 151 samples including 107 breast lesions and 44 normal controls. Dendrogram generated using ‘‘Heatmap’’ function in R with default settings.

Major: Definitely a larger sample size and more validation studies will be needed to support this method. Did authors try RNAseq instead of microarray? They need to address the fact that microarray has a limitation in which it does not allow accurate assessment of low signal intensities. And may also give background hybridization. Also the authors need to provide information on the ER, HER2 status of the lesions if available.

Response: Thanks for this constructive suggestion. Certainly RNAseq is a new and powerful tool for transcriptomic study, especially suitable for exploring unknown transcript fragments and RNA sequence variation. However, Affymetrix microarray has also been proven to be a robust and accurate technology for gene expression profiling studies. The goal of our study is to develop a technology that can exploit gene expression signatures characteristic of various breast lesion types. In previous publications, we identified a series of gene biomarkers for cancer (see references 17,31,32) that were characterized using microarray analysis and were further confirmed by RT-PCR. Thus we think Affymetrix microarray analysis is a reliable method to study gene expression signatures, as in this study. 

We have added to the manuscript, on page 22: 

“RNA sequencing (RNAseq) has been proven an efficient tool for transcriptome analysis, especially for exploring expression signatures of unknown transcript fragments and revealing the signaling pathways beneath. An interesting subject for future study would be to compare the variations in gene expression signatures between RNAseq and the microarray method.”

The treatment of data for Affymetrix microarray hybridization has been described in detail in our previous report (Chao S, Liew CC. (2015). "Mining the Dynamic Genome: A Method for Identifying Multiple Disease Signatures Using Quantitative RNA Expression Analysis of a Single Blood Sample." Microarrays 4: 671-689.). We have added this as a reference (#23) in Materials and Methods section on page 7. 

Here for your reviewers’ information is a link to this article: [https://www.ncbi.nlm.nih.gov/pmc/articles/PMC4996407/]

ER and HER2 information from patients was not completed and was unavailable, as only some patients received ER/HER2 examination in this study. 

Reviewer #2: The authors have described a ten-gene panel signature that can be used as a predictive model to detect high risk breast lesions as well as breast cancer. The authors have performed the research very comprehensively and all the analyses performed is scientifically sound. While I do believe the research performed is of significant importance in the field, I do request the authors to address a few concerns before the manuscript can be accepted for publication:

1) On page 9, the authors mention that patients with breast cancer were older than the controls as well as the ones with high-risk lesions. How do the authors know that the ten gene panel signature isn't simply reflective of aging and is directly related to the process of carcinogenesis.

Response: Certainly a larger sample size and more validation studies will be needed to support the findings described in our study. In a future report, we would like to enroll more healthy controls, more women with high risk breast lesions and more breast cancer patients with better-matched age distributions.

We have therefore added on pp. 21/22: 

“…our samples show some age variation among the healthy controls, the women with high risk lesions and the breast cancer patients. Although we have shown in our discussion that the gene biomarkers we identified are related more to the process of carcinogenesis than to aging, we intend to confirm the effectiveness of our data mining method in further studies, using a larger sample size with age-matched patients.

2) Did the author check for the known drivers of cancer and in particular breast cancer such as BRCA1,BRCA2, p53 and such other genes? Or was the analysis done in a way to find out the other genes beyond the ones that were already identified in the literature?

Response: This is an interesting question for further study. The mutations of BRCA1, BRCA2 and p53 genes have been widely observed in breast cancer tissue cells and are expected to play important roles in carcinogenesis. However in the present study, we sought to identify an altered expression signature as a characteristic of blood cells rather than of cancer tissue cells. We thus did not expect to find correlations in gene expression alterations between blood cells and breast cancer cells. By analyzing the gene expression profiles of peripheral blood cells, we found that those driver genes of cancer cells such as BRCA1, BRCA2, p53 and other genes did not exhibit significant variation in blood cells as compared to healthy controls （added on Page 21）. However, the gene biomarkers we identified in the blood-based transcriptome are shown to be related to the process of carcinogenesis in our bio-information analysis, as discussed in detail in Discussion section on pages 18-21. Further study to identify the signaling pathways of blood cells and their interaction with cancer cells is warranted to better understand the roles of blood cells in the carcinogenesis process.

We have added to our manuscript on p. 21:

“Mutations and abnormalities in the expression of BRCA1, BRCA2 and p53 genes have been widely observed in breast cancer tissue cells and are thought to play important roles as driver genes in carcinogenesis. In our study, however, BRCA1, BRCA2, p53 and other driver genes did not exhibit significant variations in expression between breast lesions and healthy controls in peripheral blood cells. This difference might be attributed to differences in biological functions between blood cells and tissue cells in the process of carcinogenesis. Further study is required in order to identify the signaling pathways of blood cells and their interaction with cancer cells to better understand the roles of blood cells in carcinogenesis.”

3) Is there a way to follow up to see if the 3 'false-positive' normal samples in their test set go on to develop high risk breast lesions and therefore check if the ten-gene panel signature is actually predictive of the early stage carcinogenesis/development of high-risk lesions?

Response: We have added on p.17:

“We are planning to follow over the next 3-5 years those healthy controls who presented with false-positive results, in order to determine whether some of them will go on to develop high-risk lesions or breast cancer.”

4) While the premise of the project and that of the manuscript is to discover a ten-gene signature panel that is predictive of developing high risk breast lesions, all the samples analyzed are grade 3 or 4 (BI-RADS). Can authors actually test if this panel is actually predictive of development of high-risk lesions by testing their gene signature panel in the grade I or grade II patients?

Response: Due to the limited sample size in the present study, the number of grade I or grade II patients is too small to generate a reliable predictive result. In a future study, we plan to recruit more patients with grade I or grade II to further validate the performance of this ten-gene panel in early detection of high-risk lesions and breast cancer.

5) Have the authors tested to see if among the 147 protein interacting partners, the transcripts of the interacting partners were also altered in their RNAseq analyses? This is only to understand if multiple genes in the pathways described in the manuscript have been altered, and therefore narrow down a potential pathway of interest for further studies?

Response: Thanks for this constructive suggestion. In the present study, we aim to identify blood-based genomic signatures that discriminate high-risk lesion and breast cancer from healthy controls using microarray analysis. As the reviewer suggests, RNAseq is a new and powerful tool for transcriptomic study, and is especially suitable for identifying unknown transcript fragments and exploring their signaling pathways. We would like to use RNAseq method to analyze and confirm potential pathways of interesting blood transcripts in a future study. 

We have added to the manuscript, on page 22: 

“RNA sequencing (RNAseq) has been proven an efficient tool for transcriptome analysis, especially for exploring expression signatures of unknown transcript fragments and revealing the signaling pathways beneath. An interesting subject for future study would be to compare the variations in gene expression signatures between RNAseq and the microarray method.”

Once these comments are addressed, the manuscript can be considered for publication.

Reviewer #3: Comments:

1. It is not clear from the description about the parameters used to identify the 10 candidate genes.

Response: the data mining method for blood-based genomic signatures has been described in our previous reports ([17,31,32]). To clarify this in the current manuscript, we have added a new reference to a previous publication that has described in detail the data mining process for blood-based signature identification (Samuel Chao, C. C., Choong-Chin Liew (2015). "Mining the Dynamic Genome: A Method for Identifying Multiple Disease Signatures Using Quantitative RNA Expression Analysis of a Single Blood Sample." Microarrays 4: 671-689.). This is added in Materials and Methods section on page 8.

We hope that the reviewer will find our methods to be robust and well-validated. Here for the reviewer’s information is a link to this article: [https://www.ncbi.nlm.nih.gov/pmc/articles/PMC4996407/]

2. It would be interesting to know the comparison of the various gene expression profile between the high risk benign breast lesion vs malignant breast lesions

Response: This is an interesting suggestion for further study. In the present study, we expected to develop blood-based genomic signatures to discriminate high-risk lesions and breast cancer from healthy controls. In a future study, we would like to explore the evolution from high risk benign lesion to malignant breast carcinoma by comparing variations between high risk benign breast lesion and malignant breast lesions. 

3. Since the sample size is small it is important to use an age matched control.

Response: we have added our response to page 22:

“…our samples show some age variation among the healthy controls, the women with high risk lesions and the breast cancer patients. Although we have shown in our discussion that the gene biomarkers we identified are related more to the process of carcinogenesis than to aging, we intend to confirm the effectiveness of our data mining method in further studies, using a larger sample size with age-matched patients.”

4. The false positivity observed in 3 of the 19 controls of the predictive model is not justified despite of small sample size

Response: We have added to page 17:

“We are planning to follow those healthy controls who presented with false-positive results over the next 3-5 years, in order to determine whether some of them will go on to develop high-risk lesions or breast cancer.”

5. Significant figures should be considered while calculating Fold change.

Response: The fold changes of the interesting 10 genes are listed in Table 3. Variation in blood gene expression profiles between cancer samples and heathy samples is usually not as significant as variations found between cancer cells and surrounding healthy tissue cells. This issue makes blood-based biomarker screening more difficult. To overcome these challenges we have developed an effective strategy for identifying blood based genomic signatures as referenced above in Question 1: https://www.ncbi.nlm.nih.gov/pmc/articles/PMC4996407/

Table 3 Candidate biomarkers for distinguishing breast lesions from controls

Probe set ID Gene Symbol Gene Title Fold Change Regulation

202887_s_at DDIT4 DNA damage inducible transcript 4 2.0014469 up

214953_s_at APP amyloid beta (A4) precursor protein 1.97339 up

214119_s_at FKBP1A FK506 binding protein 1A 1.8358978 up

202876_s_at PBX2 pre-B-cell leukemia homeobox 2 1.7287292 up

200693_at YWHAQ tyrosine 3-monooxygenase/ tryptophan 5-monooxygenase activation protein, theta 1.0801506 up

217317_s_at HERC2P2 hect domain and RLD 2 pseudogene 2 -1.2864129 down

208835_s_at LUC7L3 LUC7-like 3 pre-mRNA splicing factor -1.3374902 down

201296_s_at WSB1 WD repeat and SOCS box containing 1 -1.350631 down

201101_s_at BCLAF1 BCL2-associated transcription factor 1 -1.4449192 down

1554178_a_at FAM126B family with sequence similarity 126, member B -1.4458523 down

6. PLOS authors have the option to publish the peer review history of their article (what does this mean?). If published, this will include your full peer review and any attached files.

Do you want your identity to be public for this peer review? For information about this choice, including consent withdrawal, please see our Privacy Policy.

Reviewer #1: No

Reviewer #2: No

Reviewer #3: No

While revising your submission, please upload your figure files to the Preflight Analysis and Conversion Engine (PACE) digital diagnostic tool,https://pacev2.apexcovantage.com/. PACE helps ensure that figures meet PLOS requirements. To use PACE, you must first register as a user. Registration is free. Then, login and navigate to the UPLOAD tab, where you will find detailed instructions on how to use the tool. If you encounter any issues or have any questions when using PACE, please email us at figures@plos.org. Please note that Supporting Information files do not need this step.

---

## [Decision Letter · Decision Letter 1]

17 Mar 2020

PONE-D-19-31191R1

Peripheral blood transcriptome identifies high-risk benign and malignant breast lesions

PLOS ONE

Dear Dr Liew,

Thank you for submitting your manuscript to PLOS ONE. After careful consideration, we feel that it has merit but does not fully meet PLOS ONE’s publication criteria as it currently stands. Therefore, we invite you to submit a revised version of the manuscript that addresses the points raised during the review process.

We would appreciate receiving your revised manuscript by May 01 2020 11:59PM. To enhance the reproducibility of your results, we recommend that if applicable you deposit your laboratory protocols in protocols.io, where a protocol can be assigned its own identifier (DOI) such that it can be cited independently in the future. For instructions see: http://journals.plos.org/plosone/s/submission-guidelines#loc-laboratory-protocols

We look forward to receiving your revised manuscript.

Kind regards,

Sumitra Deb, PhD

Academic Editor

PLOS ONE

Reviewers' comments:

Reviewer's Responses to Questions

**Comments to the Author**

1. If the authors have adequately addressed your comments raised in a previous round of review and you feel that this manuscript is now acceptable for publication, you may indicate that here to bypass the “Comments to the Author” section, enter your conflict of interest statement in the “Confidential to Editor” section, and submit your "Accept" recommendation.

Reviewer #1: All comments have been addressed

Reviewer #2: All comments have been addressed

Reviewer #3: All comments have been addressed

2. Is the manuscript technically sound, and do the data support the conclusions?

Reviewer #1: (No Response)

Reviewer #2: Partly

Reviewer #3: Yes

3. Has the statistical analysis been performed appropriately and rigorously? 

Reviewer #1: (No Response)

Reviewer #2: Yes

Reviewer #3: Yes

4. Have the authors made all data underlying the findings in their manuscript fully available?

Reviewer #1: (No Response)

Reviewer #2: Yes

Reviewer #3: Yes

5. Is the manuscript presented in an intelligible fashion and written in standard English?

Reviewer #1: (No Response)

Reviewer #2: Yes

Reviewer #3: Yes

6. Review Comments to the Author

Reviewer #1: (No Response)

Reviewer #2: I believe while the authors have addressed all the comments, some of them have been loosely addressed. While most of the ones are fairly minor, there are three comments that still concern me which certainly affect the validity of the studies.

1) The samples in the two groups are not age matched. In response to this comment, the author's have added this in text of the manuscript '“…our samples show some age variation among the healthy controls, the women with high risk lesions and the breast cancer patients. Although we have shown in our discussion that the gene biomarkers we identified are related more to the process of

carcinogenesis than to aging, we intend to confirm the effectiveness of our data mining

method in further studies, using a larger sample size with age-matched patients'. I believe this as a bit of ''hand-waving'' by saying that the genes biomarkers are more related to carcinogenesis than to aging. In my opinion, while they are related to carcinogenesis, they could still be related to aging and if so the predictive value of these genes would be fairly limited as with a larger sample size they may also see a significant increase in false positives.

2) In their analysis, they had 3 normal samples show up as a false-positive. While the authors state that they plan to follow this patient for 3-5 years, there is no way to ensure this would happen for sure and therefore not predictive of early stage carcinogenesis and breast lesions as the authors state.

3) It is still surprising to me that known drivers of carcinogenesis such as p53, BRCA1, BRCA2 doe not show up in their study. While authors state 'This difference might be attributed to differences in biological functions between blood cells and tissue cells in the process of carcinogenesis.', I disagree with this argument. The assumption behind assessing peripheral gene signature is that blood contains tumor cells that are now in circulation after metastasis from the primary tumor. With this rationale, some of the known driver's should have been detected.

Reviewer #3: The authors have carefully addressed the comments to the reviewers. They have also added additional references to describe the methods used in the study . However, some comments can be addressed only by further study and using bigger sample size.

I believe the authors would consider the comments suggested by the reviewers and address them in their future studies.

7. PLOS authors have the option to publish the peer review history of their article (what does this mean?). If published, this will include your full peer review and any attached files.

Reviewer #1: No

Reviewer #2: No

Reviewer #3: No

---

## [Author Response · Author response to Decision Letter 1]

5 Apr 2020

Response to Reviewers

Reviewers' comments:

Reviewer's Responses to Questions

Comments to the Author

1. If the authors have adequately addressed your comments raised in a previous round of review and you feel that this manuscript is now acceptable for publication, you may indicate that here to bypass the “Comments to the Author” section, enter your conflict of interest statement in the “Confidential to Editor” section, and submit your "Accept" recommendation.

Reviewer #1: All comments have been addressed

Reviewer #2: All comments have been addressed

Reviewer #3: All comments have been addressed

2. Is the manuscript technically sound, and do the data support the conclusions?

Reviewer #1: (No Response)

Reviewer #2: Partly

Reviewer #3: Yes

3. Has the statistical analysis been performed appropriately and rigorously? 

Reviewer #1: (No Response)

Reviewer #2: Yes

Reviewer #3: Yes

4. Have the authors made all data underlying the findings in their manuscript fully available?

Reviewer #1: (No Response)

Reviewer #2: Yes

Reviewer #3: Yes

5. Is the manuscript presented in an intelligible fashion and written in standard English?

Reviewer #1: (No Response)

Reviewer #2: Yes

Reviewer #3: Yes

6. Review Comments to the Author

Reviewer #1: (No Response)

Reviewer #2: I believe while the authors have addressed all the comments, some of them have been loosely addressed. While most of the ones are fairly minor, there are three comments that still concern me which certainly affect the validity of the studies.

1) The samples in the two groups are not age matched. In response to this comment, the author's have added this in text of the manuscript '“…our samples show some age variation among the healthy controls, the women with high risk lesions and the breast cancer patients. Although we have shown in our discussion that the gene biomarkers we identified are related more to the process of carcinogenesis than to aging, we intend to confirm the effectiveness of our data mining method in further studies, using a larger sample size with age-matched patients'. I believe this as a bit of ''hand-waving'' by saying that the genes biomarkers are more related to carcinogenesis than to aging. In my opinion, while they are related to carcinogenesis, they could still be related to aging and if so the predictive value of these genes would be fairly limited as with a larger sample size they may also see a significant increase in false positives.

Response: Thanks for the reviewer’s valuable comments. Age has been regarded as an important risk factor for cancer, as the incidence of most cancers increases with age. In this study, which is restricted by a limited sample size, we tried to optimize the algorithm to eliminate the interference of age factors as much as possible. However, as the reviewer mentioned, it is hard to confirm that the biomarkers derived are completely unrelated to age. We also state this as a limitation in the Discussion section on page 22. In further study, we plan to validate the biomarkers and the algorithm using a larger sample size with age-matched patients. 

2) In their analysis, they had 3 normal samples show up as a false-positive. While the authors state that they plan to follow this patient for 3-5 years, there is no way to ensure this would happen for sure and therefore not predictive of early stage carcinogenesis and breast lesions as the authors state.

Response: We would like to confirm whether those 3 false positive samples are true negative samples by following them up in the next few years. For us, the purpose of this study is to develop a blood test to predict early stage carcinoma and breast lesions. Thus, it would be interesting and valuable to our future research to confirm whether the biomarkers identified from this retrospective study are in fact effective in predicting early stage breast lesions. We expect to explore this in prospective studies of breast lesions in the future. We also state this aim in our Discussion section on Page 17.

3) It is still surprising to me that known drivers of carcinogenesis such as p53, BRCA1, BRCA2 does not show up in their study. While authors state 'This difference might be attributed to differences in biological functions between blood cells and tissue cells in the process of carcinogenesis.', I disagree with this argument. The assumption behind assessing peripheral gene signature is that blood contains tumor cells that are now in circulation after metastasis from the primary tumor. With this rationale, some of the known driver's should have been detected.

Response: The gene mutations related to carcinogenesis, such as p53, BRCA1 / BRCA2, have been widely observed in breast tumor cells; however they have not been identified in our study. There are several possible reasons for this. Although tumor cells could be released into a patient’s peripheral blood, the proportion of such cells as compared with white blood cells would be very low, even for patients with advanced disease. White blood cells predominate in the cell spectrum of peripheral blood, and therefore blood gene expression signatures would largely reflect these abundant blood white cells rather than the rare circulating tumor cells. In addition, as blood white cells and tumor cells play different biological roles in the process of carcinogenesis their gene expression profiles also differ. Gene expression variations in blood white cells, for example, more likely reflect interactions between the immune system and the tumor rather than reflecting intrinsic changes within the tumor cells themselves. These differences might be an important reason why the driver genes that have been observed in tumor cells did not show abnormal signals in the gene expression profile of peripheral blood in this study. 

We have added this in the Discussion section on Pages 21-22.

Reviewer #3: The authors have carefully addressed the comments to the reviewers. They have also added additional references to describe the methods used in the study . However, some comments can be addressed only by further study and using bigger sample size.

I believe the authors would consider the comments suggested by the reviewers and address them in their future studies.

Response: on behalf of all authors, we appreciate the reviewers’ valuable suggestions very much and would like to consider them seriously in our future studies.

7. PLOS authors have the option to publish the peer review history of their article (what does this mean?). If published, this will include your full peer review and any attached files.

Do you want your identity to be public for this peer review? For information about this choice, including consent withdrawal, please see our Privacy Policy.

Reviewer #1: No

Reviewer #2: No

Reviewer #3: No

---

## [Editor Report · Decision Letter 2]

12 May 2020

Peripheral blood transcriptome identifies high-risk benign and malignant breast lesions

PONE-D-19-31191R2

Dear Dr. Liew,

We are pleased to inform you that your manuscript has been judged scientifically suitable for publication and will be formally accepted for publication once it complies with all outstanding technical requirements.

With kind regards,

Sumitra Deb, PhD

Academic Editor

PLOS ONE
---

## [Editor Report · Acceptance letter]

26 May 2020

PONE-D-19-31191R2 

Peripheral blood transcriptome identifies high-risk benign and malignant breast lesions 

Dear Dr. Liew:

I am pleased to inform you that your manuscript has been deemed suitable for publication in PLOS ONE. Congratulations! Your manuscript is now with our production department. 

With kind regards,

on behalf of

Dr. Sumitra Deb 

Academic Editor

PLOS ONE